# Active Learning is a Strong Baseline for Data Subset Selection

**Dongmin Park**[1]*, **Dimitris Papailiopoulos**[2], **Kangwook Lee**[2,3]†
[1]KAIST, [2]University of Wisconsin-Madison, [3]KRAFTON, Inc.
dongminpark@kaist.ac.kr, dimitris@papail.io, kangwook.lee@wisc.edu

## Abstract

Data subset selection refers to the process of finding a small subset of training data such that the predictive performance of a classifier trained on it is close to that of a classifier trained on the full training data. A variety of sophisticated algorithms have been proposed specifically for data subset selection. A closely related problem is the active learning problem developed for semi-supervised learning. The key step of active learning is to identify an important subset of unlabeled data by making use of the currently available labeled data. This paper starts with a simple observation – one can apply any off-the-shelf active learning algorithm in the context of data subset selection. The idea is very simple – we pick a small random subset of data and pretend as if this random subset is the only labeled data, and the rest is not labeled. By pretending so, one can simply apply any off-the-shelf active learning algorithm. After each step of sample selection, we can "reveal" the label of the selected samples (as if we label the chosen samples in the original active learning scenario) and continue running the algorithm until one reaches the desired subset size. We observe that surprisingly, this active learning-based algorithm outperforms all the current data subset selection algorithms on the benchmark tasks. We also perform a simple controlled experiment to understand better why this approach works well. As a result, we find that it is crucial to find a balance between easy-to-classify and hard-to-classify examples when selecting a subset.

## 1 Introduction

Modern deep learning have achieved unprecedented success by leveraging hyper-scale networks trained on *ever-larger* datasets, *e.g.*, GPT-3 [1], CLIP [2], and ViT [3]. However, with such a massive amount of data, practitioners often suffer from huge computational costs for model training, hyper-parameter tuning, and model architecture search. For example, training GPT-3 on 45 Tera-bytes of text data takes weeks or months even with intense GPU parallelization [4]. Reducing these computational costs is of vital important because it can accelerate model development cycles, reduce the energy consumption (*e.g.*, CO2 emission) [5], and even facilitate democratization of AI [6]. In this regard, many attempts have been devoted to prune the training dataset to enables efficient learning.

Data subset selection aims to *reduce* the size of training set by finding a core subset that generalizes on par with the full training set [7, 8]. Previous studies try to sort and select some fraction of training examples according to their difficulty for model training under the assumption that hard examples are more helpful for generalization. To do so, various score functions have been proposed, *e.g.*, uncertainty-based [9], loss-based [10, 6, 11], geometry-based [12, 13, 14], and gradient-based [15, 16, 17] scores. The prior art has proved their effectiveness by comparing the predictive performance of the model trained on the selected subsets and that of the model trained on the full data.

---

*This work was done when Dongmin Park was a research intern at KRAFTON, Inc.
†Corresponding author.

Has it Trained Yet? Workshop at the Conference on Neural Information Processing Systems (NeurIPS 2022).

A closely related, but different, problem is active learning (AL). AL was originally developed for the following semi-supervised learning setting. One is given with small labeled data and large unlabeled data. The given labeled data set is not large enough, so one would like to obtain a larger labeled dataset by labeling the unlabeled data. However, not all unlabeled examples are equal, *i.e.*, some of them will be more useful than the others once labeled. Since labeling is costly and time-consuming, it is critical to identify which examples are more useful than the others so that we can prioritize labeling important ones given a labeling budget. AL is a principled approach to solving this problem – see [18] for a detailed survey of AL for deep learning.

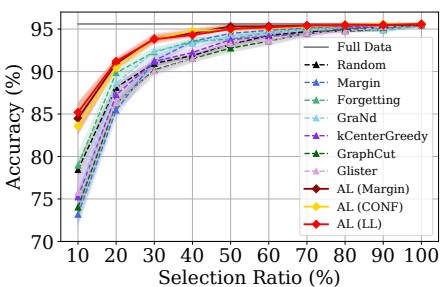

Figure 1: Test accuracy of existing data subset selection methods and AL baselines on CIFAR10. While AL baselines do not use labels when selecting samples, they beat every subset selection method.

While AL is developed with semi-supervised learning problems in mind, it is inherently related to the data subset selection problem in that both are concerned with finding important subsets. In fact, it is straightforward to observe that any AL algorithm can be modified so that it can be used for the data subset problem. The idea is strikingly simple – we start with a small random subset of data and pretend as if it is the only labeled data; train a classifier on the selected subset; augment the subset with a number of training examples selected by an AL algorithm; and repeat the training and augmenting steps until selecting the target number of subset examples.

In this paper, we show that AL can be a strong baseline for data subset selection. As shown in Figure 1, we observe that a simple AL algorithm, such as AL (Margin) [19], outperforms every state-of-the-art data subset selection algorithm in terms of test accuracy regardless of the selection ratios. We demonstrate the superiority of this baseline using various architectures and datasets including CIFAR10, CIFAR100, and a subset of ImageNet.

This is very surprising in that AL does *not* use the ground truth labels when selecting examples whereas the subset selection does; AL uses less information than subset selection but achieves better performance. We also perform a simple controlled experiment to better understand why this baseline works well. As a result, we find that it is crucial to find a balance between easy-to-classify and hard-to-classify examples when selecting a subset. Accordingly, we contend that it now becomes an open problem to develop a new subset selection algorithm that outperforms our simple AL baseline.

## 2 Preliminary

### 2.1 Data Subset Selection

**Problem Setup.** Let $\mathcal{D} = \{(x_i, y_i)\}_{i=1}^{m}$ be a given training set obtained from a joint distribution $\mathcal{X} \times \mathcal{Y}$, where $m$ denotes the number of training examples, $\mathcal{X}$ and $\mathcal{Y}$ denote the input space and label space, respectively. Data subset selection aims to identify the most informative subset $\mathcal{S} \subset \mathcal{D}$, so that the model $\theta_{\mathcal{S}}$ trained on $\mathcal{S}$ can maintain possibly close generalization performance to the model $\theta_{\mathcal{D}}$ trained on the entire training set $\mathcal{D}$.

**Related Work.** Coleman et al. [9] shows that the uncertainty-based scores, e.g., Confidence [20], Margin [19], and Entropy [21], can be effective metrics for subset selection in that selecting lower confident examples is more helpful for model generalization than selecting higher confident ones. Some work used the geometric distance in the feature space to avoid selecting examples with redundant information. Herding [12] incrementally extends the selected coreset by greedily adding a data example which can minimize the distance between the center of the coreset and that of the original training set. kCenterGreedy [14] selects $k$ examples that maximize the distance coverage on the entire unlabeled data. Recently, many approaches try to directly exploit the components of deep learning with given ground-truth labels. Forgetting [10] selects examples that are easy to be forgotten by the classifier, and it finds such samples by counting how frequently predicted label changes during several warm-up training epochs. GraNd [6] uses the average norm of gradient vector to measure the contribution of each examples for minimizing the training loss. GradMatch [16] and CRAIG [15] try to find an optimal coreset that its gradient can be matched with the gradient of full training set. Glister [17] introduces a bi-level optimization framework that the outer loop is

for selecting the coreset which can be solved by a greedy algorithm. Submodular functions, such as Graph Cut, Facility Location, and Log Determinant, which measure the diversity of information, have also been shown to be useful for data subset selection [22].

## 2.2 Active Learning

**Problem Setup.** Let $\mathcal{U} = \{x_i\}_{i=1}^m$ be a given unlabeled set, where $x_i \sim \mathcal{X}$. Active learning aims to select the most informative examples from $\mathcal{U}$ and turn them into the labeled set $\mathcal{L} \subset \mathcal{D}$ by assigning the ground-truth label $y$ on each example $x$ with a human oracle. The AL process—selecting, labeling, and training—repeats for multiple rounds, say $r$ rounds, until attaining the target number of examples. Usually, the initial labeled set $\mathcal{L}_0$ are selected randomly. The labeled set $\mathcal{L}$ extends gradually by adding a newly selected set for each rounds, *i.e.*, $\mathcal{L} \leftarrow \mathcal{L} \cup \mathcal{L}_r$.

**Related Work.** Numerous AL scores for measuring informativeness of examples without given the ground-truth labels have been proposed [18, 23]. One typical type is uncertainty-based score, such as soft-max Confidence [20], Margin [19], and Entropy [21], which is calculated on the final prediction probability of the model. Some approaches measures uncertainty by using the Monte Carlo Dropout [24] throughout multiple forward passes [25, 26]. LL [27] uses the predicted loss obtained by a small loss prediction module which is jointly learned with the target model. Another popular type of methods is diversity-based sampling which incorporates a clustering [28, 29, 30] or greedy selection algorithm [14] to select diverse examples that well-represent the entire data distribution.

## 2.3 Applying Active Learning Algorithm to Data Subset Selection

Given a fully labeled dataset $\mathcal{D}$, one can pretend that label information is not given and view it as $\mathcal{U}$. Then, one can choose a small labeled random subset $\mathcal{L}_0$ from $\mathcal{D}$. One can obtain the indices of chosen samples by running one step of an active learning algorithm on $\mathcal{L}_0$. By retrieving the labels of these samples from $\mathcal{D}$, one can simulate the oracle labeling process, hence obtaining $\mathcal{L}_1$. One can repeat this process until this process reaches the desired subset size.

# 3 Experiment

## 3.1 Experiment Setting

**Datasets.** We perform the data subset selection and active learning on three datasets; CIFAR10 [31], CIFAR100 [31], and ImageNet-30 [32], a subset of ImageNet [33] containing 30 classes.

**Algorithms.** We compare with a random selection, *six* subset selection algorithms, including Margin [19], Forgetting [10], GraNd [6], kCenterGreedy [14], GraphCut [22], and Glister [17], and *three* AL algorithms, AL (Margin) [19], AL (Conf) [20], and AL (LL) [27].

**Implementation Details.** For all datasets, we conduct the performance comparison with the selection ratios of $\{10\%, 20\%, \ldots, 90\%\}$. For data subset selection methods, following the prior work [7], a warm-up training of 10 epochs on the entire training set is preceded to calculate the difficulty scores. For AL baselines, we randomly select $2\%$ of training examples as the initial labeled set, and add $2\%$ of training examples at every AL round. We run every experiment three times and report the average of the best accuracy. More implementation details with training configurations can be found in Appendix A. The code is available at `https://github.com/dongmean/AL_vs_SubsetSelection`.

## 3.2 Performance Comparison.

Table 1 shows the detailed performance of data subset selection methods and AL baselines over various selection ratios on CIFAR10, CIFAR100, and ImageNet30 with ResNet-18. See Appendix A for the result with VGG architecture. The overall performance curves for CIFAR10 are illustrated in Figure 1 and those for CIFAR100 and ImageNet30 are depicted in Figure 3 in Appendix B.

In general, AL baselines outperform all subset selection methods over the most selection ratios. Specifically, AL (Margin) succeeds to maintain the full test accuracy within an error of $0.5\%$ until the selection ratio of $50\%$ for CIFAR10, $80\%$ for CIFAR100, and $70\%$ for ImageNet30. On the other hand, the performance of subset selection methods drastically degrade when the selection ratio is very low and is sometimes even worse than Random, *e.g.*, for CIFAR10, every subset selection method except Forgetting is worse than Random at the selection ratios of $10\%$ and $20\%$.

Table 1: Performance comparison of data subset selection and AL methods on CIFAR10, CIFAR100, and ImageNet30. We train ResNet-18 from scratch on the selected set. The best results are in bold.

| Datasets | Select Ratios | 10% | 20% | 30% | 40% | 50% | 60% | 70% | 80% | 90% | 100% |
|---|---|---|---|---|---|---|---|---|---|---|---|
| CIFAR10 | Random | 78.4±0.9 | 88.1±0.5 | 91.0±0.3 | 91.9±0.2 | 93.2±0.3 | 94.3±0.2 | 94.7±0.2 | 94.8±0.1 | 95.1±0.2 | 95.5±0.2 |
| | Margin | 73.2±1.3 | 85.5±0.9 | 91.3±0.5 | 93.6±0.3 | 94.5±0.2 | 94.9±0.3 | 95.1±0.1 | 95.4±0.2 | **95.5±0.1** | 95.5±0.2 |
| | Forgetting | 79.0±1.0 | 89.8±0.9 | 92.3±0.4 | 93.6±0.4 | 93.8±0.3 | 94.6±0.4 | 95.0±0.2 | 95.1±0.2 | 95.4±0.2 | 95.5±0.2 |
| | GraNd | 75.4±1.2 | 88.6±0.6 | 92.4±0.4 | 93.3±0.5 | 94.2±0.4 | 94.6±0.3 | 95.0±0.2 | 95.1±0.2 | **95.5±0.2** | 95.5±0.2 |
| | kCentGreedy | 75.2±1.7 | 87.3±1.0 | 91.2±0.6 | 92.2±0.5 | 93.8±0.5 | 94.2±0.4 | 94.4±0.3 | 95.1±0.2 | **95.5±0.2** | 95.5±0.2 |
| | GraphCut | 74.0±1.5 | 86.3±0.9 | 90.2±0.5 | 91.5±0.4 | 93.8±0.6 | 94.2±0.4 | 94.4±0.3 | 95.1±0.2 | **95.5±0.2** | 95.5±0.2 |
| | Glister | 75.7±1.0 | 86.3±0.9 | 90.1±0.7 | 91.5±0.5 | 93.3±0.6 | 93.6±0.6 | 94.5±0.4 | 94.8±0.3 | 95.2±0.2 | 95.5±0.2 |
| | AL(Margin) | 84.5±0.7 | 91.0±0.5 | **93.9±0.4** | 94.5±0.3 | **95.3±0.2** | 95.3±0.2 | 95.4±0.2 | 95.5±0.2 | 95.5±0.1 | 95.5±0.2 |
| | AL(Conf) | 83.6±0.7 | 90.5±0.4 | 93.8±0.4 | **94.8±0.3** | 95.1±0.3 | **95.3±0.2** | **95.4±0.2** | 95.4±0.2 | **95.5±0.2** | 95.5±0.2 |
| | AL(LL) | **85.0±0.9** | **91.2±0.7** | 93.8±0.6 | 94.4±0.5 | 95.0±0.4 | 95.2±0.4 | 95.4±0.3 | 95.5±0.3 | 95.5±0.2 | 95.5±0.2 |
| CIFAR100 | Random | 32.0±0.9 | 53.6±0.6 | 63.6±0.5 | 67.2±0.5 | 71.0±0.3 | 73.1±0.4 | 75.2±0.2 | 76.1±0.3 | 77.5±0.2 | 78.7±0.2 |
| | Margin | 18.7±2.1 | 38.2±1.6 | 58.1±0.8 | 65.1±0.6 | 70.1±0.5 | 73.3±0.3 | 75.4±0.3 | 76.9±0.4 | **78.5±0.2** | 78.7±0.2 |
| | Forgetting | 35.4±1.0 | 54.7±0.9 | 64.6±0.7 | 68.6±0.8 | 71.5±0.4 | 73.7±0.5 | 75.5±0.3 | 76.1±0.3 | 76.9±0.3 | 78.7±0.2 |
| | GraNd | 30.8±1.9 | 49.4±1.0 | 62.8±0.9 | 68.1±0.6 | 70.5±0.3 | 72.5±0.4 | 74.5±0.3 | 76.4±0.2 | 77.8±0.2 | 78.7±0.2 |
| | kCentGreedy | 33.9±1.5 | 56.2±0.9 | 64.5±0.6 | 69.8±0.4 | 72.1±0.5 | 74.3±0.4 | 75.8±0.3 | 77.2±0.2 | 77.8±0.2 | 78.7±0.2 |
| | GraphCut | 36.3±1.1 | 56.0±0.8 | 65.5±0.6 | 69.5±0.4 | 71.1±0.4 | 73.8±0.4 | 75.4±0.2 | 76.4±0.2 | 78.0±0.2 | 78.7±0.2 |
| | Glister | **36.4±1.0** | 55.5±1.0 | 63.9±0.8 | 69.1±0.7 | 71.2±0.6 | 73.5±0.4 | 75.0±0.3 | 76.9±0.2 | 77.6±0.2 | 78.7±0.2 |
| | AL(Margin) | 36.0±1.0 | **57.3±0.5** | **66.0±0.6** | 70.4±0.5 | 73.6±0.5 | **76.1±0.4** | 77.2±0.3 | 78.2±0.3 | **78.5±0.2** | 78.7±0.2 |
| | AL(Conf) | 36.1±1.6 | 55.7±1.0 | 65.8±0.7 | **70.6±0.5** | **73.7±0.4** | **76.1±0.5** | 77.1±0.3 | 78.0±0.2 | 78.4±0.2 | 78.7±0.2 |
| | AL(LL) | 33.1±1.9 | 55.3±1.3 | 64.9±0.8 | 70.3±0.7 | 73.1±0.5 | 75.9±0.5 | 77.0±0.3 | **78.2±0.3** | **78.5±0.2** | 78.7±0.2 |
| ImageNet30 | Random | 69.3±0.7 | 83.7±0.5 | 86.9±0.4 | 90.3±0.3 | 92.2±0.3 | 93.0±0.2 | 94.6±0.3 | 95.2±0.2 | 95.4±0.2 | 96.1±0.1 |
| | Margin | 56.9±1.1 | 77.3±0.7 | 83.7±0.5 | 90.5±0.4 | 92.9±0.2 | 94.4±0.3 | 95.1±0.2 | **95.8±0.2** | **96.0±0.1** | 96.1±0.1 |
| | Forgetting | 64.1±0.9 | 85.4±0.7 | 87.3±0.5 | 90.9±0.3 | 93.6±0.4 | 94.8±0.2 | 94.9±0.2 | 95.1±0.2 | 95.3±0.2 | 96.1±0.1 |
| | GraNd | 69.3±0.9 | 85.7±0.5 | 90.0±0.5 | 92.4±0.4 | 93.6±0.3 | 94.7±0.4 | 95.1±0.2 | 95.5±0.3 | 95.7±0.1 | 96.1±0.1 |
| | kCentGreedy | 69.7±0.9 | 84.1±0.5 | 88.9±0.4 | 91.6±0.3 | 93.4±0.2 | 94.4±0.3 | 95.1±0.2 | 95.3±0.2 | 95.6±0.2 | 96.1±0.1 |
| | GraphCut | 71.9±0.6 | 83.0±0.3 | 88.5±0.3 | 91.2±0.3 | 92.9±0.2 | 93.7±0.3 | 94.4±0.2 | 95.3±0.2 | 95.8±0.2 | 96.1±0.1 |
| | Glister | **72.4±0.7** | 82.9±0.5 | 87.0±0.4 | 91.2±0.3 | 92.7±0.3 | 93.3±0.3 | 94.2±0.2 | 95.0±0.2 | 95.8±0.2 | 96.1±0.1 |
| | AL(Margin) | 71.9±0.9 | 86.7±0.5 | 90.1±0.4 | **93.3±0.4** | 94.5±0.3 | 95.1±0.2 | **95.6±0.3** | **95.8±0.2** | **96.0±0.2** | 96.1±0.1 |
| | AL(Conf) | 70.7±1.1 | **87.0±0.5** | **90.3±0.5** | 93.1±0.4 | 94.3±0.3 | 95.1±0.2 | 95.5±0.4 | 95.7±0.2 | **96.0±0.1** | 96.1±0.1 |
| | AL(LL) | 68.4±1.5 | 85.5±0.7 | 89.3±0.6 | 93.1±0.5 | **94.7±0.2** | **95.3±0.2** | **95.6±0.3** | **95.8±0.2** | **96.0±0.2** | 96.1±0.1 |

## 3.3 Quick Analysis: Why a Simple AL Algorithm is Better than Data Subset Selection?

One main difference between data subset selection and AL is that AL starts from a small *random* initial set. We study the effect of the random set to subset selection with a simple controlled experiment.

**Experiment Setup.** We first train a warm-up classifier on the entire CIFAR10 for 10 epochs. Our goal is to compare the accuracy of the model trained on subset of two cases. For the first case, we randomly select 5,000 examples from CIFAR10 as an initial set. Then, using the warm-up classifier, we calculate the Margin score on the remaining 45,000 examples and divide them into 9 bins, *i.e.*, 5,000 examples per bin, according to the score. Next, we compose 5 subsets by only combining each odd-numbered bin with the random initial set (10,000 examples per each subset). For the second case,

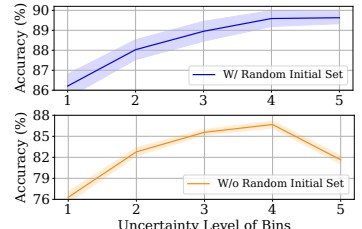

Figure 2: Effect of the random initial set on performance.

using the warm-up classifier, we calculate the Margin score on the entire 50,000 examples and divide them into 5 bins, *i.e.*, 10,000 examples per bin, according to the score.

**Result.** Figure 2 shows the performance of selected subset with or without the random initial set. With the random initial set (blue line), the performance increases according to the uncertainty level of each bin; the hardest examples are the most helpful in improving model performance. In contrast, without the random initial set (yellow line), the model performance is not the best at the bin of highest uncertainty, which can explain the drastic performance drop of subset selection methods in low selection ratio in Section 3.2. This indicates that the hardest examples give the greatest benefit to model generalization only when they are used together with easy examples. Based on this finding, we further show the data subset selection can be improved by adding two main components in AL; random initial set and multi-round data selection (the result can be found in Appendix D).

## 4 Conclusion

In this work, we showed that AL can be a strong baseline for data subset selection via extensive experimental results on multiple datasets. This is surprising as AL does not use the ground-truth label information when selecting samples. We expect that our observation calls a new attention to developing a new data subset selection algorithm that can outperform the AL baselines.

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

# Active Learning is a Strong Baseline for Data Subset Selection
## (Supplementary Material)

## A  Implementation Details

For CIFAR10 and CIFAR100 datasets, we train ResNet-18 [34] and VGG-19 [35] from scratch for 200 epochs using SGD with batch size 128, momentum 0.9, weight decay $5 \times 10^{-4}$, and initial learning rate 0.1 with the cosine decay scheduler. For data augmentation, we apply random horizontal flipping and random crop with 4-pixel padding. For ImageNet-30 dataset, we train ResNet-18 and VGG-16 from scratch for 200 epochs using SGD with batch size 128, weight decay $5 \times 10^{-4}$, and initial learning rate 0.1 with the cosine decay scheduler. For data augmentation, we resize the images to $256 \times 256$, randomly crop it to $224 \times 224$, and apply random horizontal flipping.

For implementation of all data subset selection algorithms, we use the code in DeepCore library[3]. The hyperparameters for all algorithms are favorably configured following the original papers. All algorithms are implemented with PyTorch 1.8.0 and executed on a single NVIDIA Tesla A100 GPU.

## B  Performance Curves on CIFAR100 and ImageNet30

Figure 3 illustrates the overall performance curves of data subset selection algorithms and an AL baseline on CIFAR100 and ImageNet30 with ResNet-18. Similar to the result in Figure 1 on CIFAR10, AL (Margin) outperforms all the subset selection algorithms over the most selection ratios. Among the subset selection algorithms, Margin shows the most extreme performance drop as the selection ratio becomes lower.

## C  Results with VGG Architecture

Table 2 shows the detailed performance of data subset selection and an AL baseline with VGG-19 architecture for CIFAR10 and CIFAR100 and VGG-16 architecture for ImageNet30. Similar to Section 3.2, AL (Margin) wins all data subset selection methods regardless of the selection ratios. Specifically, AL (Margin) succeeds to maintain the full test accuracy within an error of 0.5% until the fraction ratio of 50% for CIFAR10, 80% for CIFAR100, and 70% for ImageNet30. This indicates that AL is better than data subset selection is consistent across the network architectures.

## D  Effect of Random Subset and Multi-round Selection to Subset Selection

**Experiment Setup.** We make two modified versions of a data subset selection algorithm (Margin) where each of which incorporates two main components of AL; 1) random initial set, and 2) multi-round selection, respectively. For the first version, we randomly extract 2% of data examples from the entire training set and perform data subset selection with the warm-up training on the non-extracted 98% of training set. Then, when selecting the final subset, we combine the randomly extracted examples with some fraction of hardest examples, *e.g.*, when the target selection ratio is 10%, we select 8% of hardest examples from the training set and combine it with already extracted 2% random examples. For the second version, we repeatedly remove 2% of the easiest examples from the entire training set and redo the warm-up training on the remaining set whenever we remove the examples until reaching to the target selection ratio. This version also selects the examples that are less hard than the original data subset selection with one-shot warm-up training, because the accuracy of warm-up training gradually decreases as less amount of examples is remained by the repeated example removal; the uncertainty score becomes less confident. We train ResNet-18 [34] with the same training configuration in Appendix A

**Result.** Table 3 shows the performance of two modified versions of Margin. Overall, both versions outperforms the original Margin, which means balancing easy-to-classify and hard-to-classify is beneficial to data subset selection. Nevertheless, each version is not yet better than the AL (Margin), which benefits both random initial set and multi-round selection.

---

[3] https://github.com/PatrickZH/DeepCore

# E   Problem Scope

While a number of prior studies have focused on finding a subset in the early learning phase to reduce the time to train a model *once* so that avoid losing the computational gain of training on less data, we focus on the problem that we train models *multiple* times on the selected subset (hyper-parameter tuning, neural architecture search, continual learning, etc.), where the computational cost is not a primary concern.

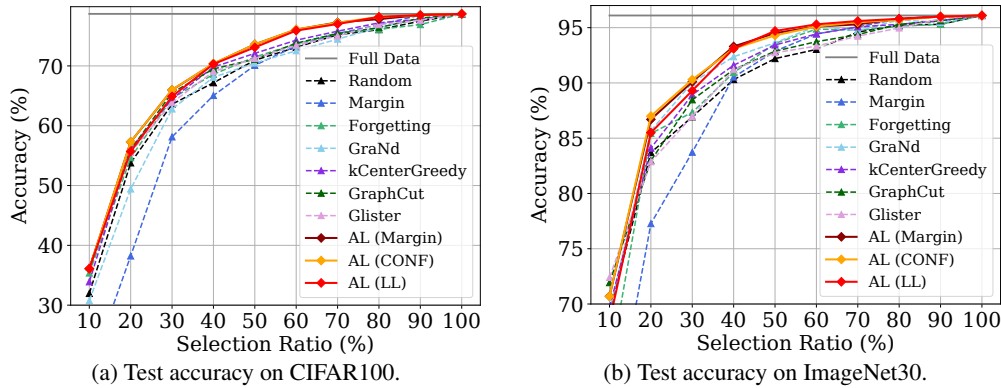

Figure 3: Performance comparison of existing data subset selection methods and AL baselines on CIFAR100 and ImageNet30 with ResNet-18.

Table 2: Performance comparison of data subset selection and AL on CIFAR10 and CIFAR100 with VGG-19, and on ImageNet30 with VGG-16. The best results are in bold.

| Datasets | Select Ratios | 10% | 20% | 30% | 40% | 50% | 60% | 70% | 80% | 90% | 100% |
|---|---|---|---|---|---|---|---|---|---|---|---|
| CIFAR10 | Random | 76.7±0.8 | 85.6±0.6 | 89.8±0.4 | 91.5±0.3 | 92.5±0.3 | 93.1±0.2 | 93.3±0.2 | 93.4±0.3 | 93.7±0.2 | 94.1±0.2 |
| | Margin | 48.3±1.5 | 81.6±0.7 | 89.0±0.4 | 91.4±0.4 | 92.9±0.3 | 93.3±0.3 | 93.5±0.2 | **93.8±0.2** | 94.0±0.1 | 94.1±0.2 |
| | Forgetting | 66.7±1.1 | 85.6±0.5 | 89.8±0.5 | 91.5±0.3 | 92.5±0.3 | 93.1±0.2 | 93.3±0.2 | 93.4±0.2 | 93.8±0.2 | 94.1±0.2 |
| | GraNd | 52.0±1.3 | 83.7±0.4 | 89.7±0.5 | 92.0±0.4 | 92.9±0.4 | 93.4±0.3 | **93.8±0.2** | **93.8±0.1** | **94.1±0.2** | 94.1±0.2 |
| | kCentGreedy | 76.8±0.8 | **86.1±0.6** | 88.7±0.4 | 90.9±0.4 | 91.8±0.3 | 92.6±0.2 | 92.9±0.3 | 93.5±0.2 | 93.8±0.2 | 94.1±0.2 |
| | GraphCut | 77.2±0.6 | 84.9±0.5 | 88.0±0.4 | 89.8±0.3 | 91.1±0.4 | 92.3±0.3 | 93.2±0.2 | 93.5±0.2 | **94.1±0.2** | 94.1±0.2 |
| | Glister | 76.7±0.7 | 85.0±0.5 | 87.9±0.4 | 90.1±0.4 | 90.9±0.3 | 91.8±0.3 | 92.3±0.3 | 93.1±0.2 | 93.5±0.2 | 94.1±0.2 |
| | AL(Margin) | **78.0±0.6** | **86.1±0.6** | **89.9±0.4** | **92.3±0.4** | **93.1±0.2** | **93.6±0.2** | **93.8±0.2** | **93.8±0.2** | **94.1±0.2** | 94.1±0.2 |
| CIFAR100 | Random | 28.3±1.2 | 48.9±1.0 | 58.0±0.7 | 62.6±0.5 | 64.8±0.5 | 67.3±0.4 | 69.2±0.3 | 70.9±0.3 | 71.9±0.3 | 73.5±0.2 |
| | Margin | 14.6±2.2 | 35.5±1.7 | 50.0±1.0 | 58.1±0.7 | 63.1±0.5 | 66.7±0.4 | 69.7±0.3 | 71.6±0.4 | 73.3±0.2 | 73.5±0.2 |
| | Forgetting | **29.9±1.9** | 52.1±1.2 | 59.0±0.9 | 63.9±0.6 | 67.1±0.5 | 68.6±0.5 | 69.6±0.4 | 71.3±0.3 | 72.5±0.2 | 73.5±0.2 |
| | GraNd | 25.7±2.0 | 47.2±1.4 | 57.2±1.1 | 63.8±0.9 | 66.6±0.6 | 68.5±0.5 | 70.2±0.3 | 71.9±0.3 | 72.8±0.2 | 73.5±0.2 |
| | kCentGreedy | 22.2±1.6 | 49.4±1.3 | 57.9±0.9 | 62.7±0.7 | 66.5±0.5 | 68.0±0.6 | 69.3±0.4 | 71.9±0.3 | 72.6±0.3 | 73.5±0.2 |
| | GraphCut | 29.9±1.5 | 49.1±1.1 | 57.1±0.8 | 62.4±0.5 | 65.7±0.6 | 68.0±0.4 | 69.2±0.4 | 70.8±0.3 | 72.5±0.2 | 73.5±0.2 |
| | Glister | 21.5±1.9 | 49.4±1.2 | 57.7±0.8 | 63.0±0.8 | 66.0±0.6 | 67.7±0.5 | 69.7±0.4 | 71.1±0.3 | 72.2±0.2 | 73.5±0.2 |
| | AL(Margin) | 28.2±1.9 | **49.6±1.0** | **59.1±0.6** | **64.6±0.5** | **69.3±0.4** | **70.1±0.4** | **71.9±0.3** | **73.0±0.2** | **73.4±0.2** | 73.5±0.2 |
| ImageNet30 | Random | **69.6±0.8** | 80.9±0.5 | 85.9±0.3 | 90.1±0.3 | 91.6±0.3 | 93.3±0.3 | 93.7±0.2 | 94.6±0.3 | 94.8±0.2 | 95.7±0.1 |
| | Margin | 53.8±1.5 | 76.3±0.8 | 84.6±0.5 | 90.8±0.5 | 93.1±0.4 | 94.2±0.4 | 95.0±0.2 | 95.2±0.3 | 95.4±0.2 | 95.7±0.1 |
| | Forgetting | 63.8±1.1 | 81.4±0.8 | 88.1±0.6 | 90.6±0.5 | 93.0±0.3 | 93.3±0.3 | 93.6±0.3 | 94.6±0.2 | 95.2±0.2 | 95.7±0.1 |
| | GraNd | 64.3±1.1 | 80.0±0.8 | 88.6±0.6 | 90.9±0.4 | 92.2±0.3 | 93.0±0.4 | 93.8±0.3 | 94.5±0.2 | 95.2±0.1 | 95.7±0.1 |
| | kCentGreedy | 66.3±1.0 | 81.3±0.7 | 88.7±0.6 | 90.4±0.4 | 91.7±0.4 | 93.3±0.3 | 93.7±0.2 | 94.4±0.2 | 94.9±0.2 | 95.7±0.1 |
| | GraphCut | 68.3±1.2 | 81.7±0.6 | 87.3±0.5 | 89.2±0.3 | 91.9±0.3 | 92.8±0.3 | 93.5±0.2 | 94.1±0.3 | 94.9±0.2 | 96.1±0.1 |
| | Glister | 69.1±0.7 | 80.8±0.5 | 87.2±0.5 | 89.6±0.4 | 91.5±0.3 | 92.8±0.3 | 93.5±0.3 | 94.3±0.2 | 94.7±0.2 | 95.7±0.1 |
| | AL(Margin) | 69.5±1.2 | **84.6±0.6** | **89.1±0.6** | **92.5±0.4** | **93.8±0.4** | **94.9±0.3** | **95.3±0.3** | **95.4±0.2** | **95.7±0.1** | 95.7±0.1 |

Table 3: Effect of incorporating random initial set and multi-round selection into data subset selection on CIFAR10 with ResNet-18.

| Select Ratio | 10% | 20% | 30% | 40% | 50% | 60% | 70% | 80% | 90% | 100% |
|---|---|---|---|---|---|---|---|---|---|---|
| Margin | 73.2±1.3 | 85.5±0.9 | 91.3±0.5 | 93.6±0.3 | 94.5±0.2 | 94.9±0.3 | 95.1±0.1 | 95.4±0.2 | **95.5±0.2** | 95.5±0.2 |
| Margin + Random Init | 81.2±1.2 | 88.4±0.7 | 92.1±0.5 | 93.9±0.4 | 94.7±0.3 | 95.1±0.2 | **95.4±0.2** | **95.5±0.2** | **95.5±0.2** | 95.5±0.2 |
| Margin + Multi Round | 80.1±1.0 | 89.2±0.5 | 93.0±0.3 | 94.3±0.4 | 94.8±0.3 | **95.3±0.3** | **95.4±0.2** | **95.5±0.2** | **95.5±0.2** | 95.5±0.2 |
| AL(Margin) | **84.5±0.7** | **91.0±0.5** | **93.9±0.4** | **94.5±0.3** | **95.3±0.2** | **95.3±0.2** | **95.4±0.2** | **95.5±0.2** | **95.5±0.1** | 95.5±0.2 |

