# OpenReview forum: "Active Learning is a Strong Baseline for Data Subset Selection"
_NeurIPS.cc/2022/Workshop/HITY — HITY Workshop NeurIPS 2022_

### Official Review · Reviewer_MFB3 · 2022-10-06
**Promising and interesting results, clear text and of relevance for this Workshop.**

**Rating:** 1
**Confidence:** 4

**Review:**

In this paper, it is shown that algorithms based on active learning perform better than recent data subset selection algorithms.  Furthermore, an attempt has been made to explain why this is the case. The results look promising, and the text is clear.

Figure 1 and Figure 2: Please provide information over how many seeds you averaged and state the uncertainty intervals. If only one measurement was done, this result may not be significant.

Section 3.3: Please explain why it is a fair and valid comparison if the one algorithm uses 18000 samples, whereas the other only uses 8000. To me, a fair comparison would have been if only a subset of the random initial set is used to keep the amount of training data comparable (i.e. 4000 random and 4000 from the corresponding bin).
Please repeat this experiment on CIFAR-100 and Imagenet-30 to strengthen your statements.

---

### Official Review · Reviewer_xAVm · 2022-10-12

**Rating:** 1
**Confidence:** 4

**Review:**

This paper provides an observation that active learning, which selects unlabeled data to be labeled by a human oracle, can be a strong baseline in data subset selection tasks. The idea is very simple: pretend that the labeled dataset $D$ is unlabeled and let active learning choose a subset of it, then simply use that subset to train the network using the corresponding labels in $D$.

I think this paper can spark interesting discussions.

---

### Decision · Program_Chairs · 2022-10-20

Accept